# A Study on Fabrication Process of Gold Microdisk Arrays by the Direct Imprinting Method Using a PET Film Mold

Potejana Potejanasak 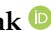

Department of Industrial Engineering, School of Engineering, University of Phayao, Phayao 56000, Thailand; potejanasak.po@up.ac.th

**Abstract:** In this study, an efficient nanofabrication process of metal microdisk arrays using direct imprinting was developed. This process was comprised of three steps; sputter etching on the quartz glass substrate, gold thin film deposition on an etched surface of a substrate, and transfer imprinting using a polyethylene terephthalate (PET) film mold on the Au thin film. A new idea to utilize a PET film mold for disk patterning by the nano transfer imprinting was examined. The PET film mold was prepared by thermally embossing the pillar pattern of a master mold on the PET film. The master mold was prepared from a silicon wafer. The PET film mold was used for transfer imprinting on a metal film deposited on a quartz substrate. The experimental results revealed that the PET film mold can effectively form gold micro-disk arrays on the Au film despite the PET film mold being softer than the Au film. This method can control the distribution and orientation of the nano-arrays on the disk. The plasmonic properties of the gold micro-disk arrays are studied and the absorbance spectrum exhibit depends on the distribution and orientation of gold micro-disk patterns. The nano-transfer imprinting technique is useful for fabricating metallic microdisk arrays on substrate as a plasmonic device.

**Keywords:** gold micro-disk arrays; transfer imprinting method; PET film mold; thermal embossing technique

## 1. Introduction

Noble metallic nanostructures exhibit unique optical properties, such as localized surface plasmon resonance (LSPR). This LSPR occurs in metallic nanoparticles when the electrons on the nanodot arrays are excited by a specific wavelength of the visible light region. The spectral characteristics of LSPR depend on the design, dimension, orientation, shape, and arrangement of the metallic nanostructures [1–3]. The LSPR resonant frequency depends on the refraction index of the surrounding environmental medium. Metal nanostructures can be used as an optical plasmonic biosensors, for example, for virus detection [4,5]. Since the position and magnitude of the extinction spectrum are sensitive to shifts in the local environment caused by the chemical molecules adsorbed onto surface of metallic nanoparticles, the nanostructure can serve as transmission-based biosensor [6]. The gold nanoplate arrays are an interesting design of plasmonic metallic nanostructures that are popularly used for biomedical applications. This LSPR-based refractive index sensing technique using gold disk arrays on a substrate enables the real-time label-free detection of cancer markers [7].

Currently, lithographic techniques are the most commonly used top-down methods in the fabrication of such ordered metallic nanostructures on a substrate. The most commonly used technique is electron beam lithography (EBL). Methods include electron sensitive resist deposition, patterning of the shadow mask by electron beam, deposition of the metal layer, and chemical lift-off processes [8–12]. Another approach, called focused ion beam etching (FIB), typically uses a gallium ion beam to sputter unwanted parts directly onto a metal layer [13,14]. While both techniques can produce nanostructures with well-controlled dimensions and arrangements, they require expensive equipment, stringent



control processes, and cannot address the problem of low throughput. Nanoimprint lithography (NIL) is an alternative top-down approach that has the advantage of producing high resolution of nanostructures. However, the nanopatterns on the mold required in the NIL technique are usually fabricated by conventional lithography technologies [15–18]. Therefore, NIL cannot overcome the disadvantages of conventional lithography techniques.

On the other hand, bottom-up approaches are expected to be high-throughput, low-cost, and less stringent technologies through self-organization processes. For example, anodic aluminum oxide (AAO) is extensively studied [19], although this process is used to fabricate large-area nanostructures with high-resolution features. However, the morphology of anodic porous alumina is limited by the electrolyte during the anodization process. The dimension and geometrical arrangement of the hexagonal nano-hole array is constrained by the conditions of self-assembly. Another approach for the fabrication of a nanodisk array is based on the thermal dewetting process of thin films on a substrate. The thermal dewetting method of metal thin films provides a simple, low-cost, and high-throughput method for fabricating nanostructures on a substrate. However, this method has the disadvantage of random distribution of the nanodisk dimension and orientation [20–22]. Although there are many studies on nanofabrication of metallic nanostructure arrays on a substrate for optical biosensing, there are still some remaining fabrication gaps that deserve attention.

To overcome these fabrication limitations, this research work develops new efficient nanofabrication methods for metallic nanostructures on substrates. The transfer imprinting method using a polymer film mold was experimentally investigated. Experimental testing for micro-hold grid patterning was investigated as an alternative to the conventional methods. In this study, an optical film of polyethylene terephthalate (PET) was used as a microform to fabricate gold micro-disk arrays on the deposited metal thin film on the substrate. The PET film has the potential to be very cheap and easy to fabricate. However, it was found that the dimensions of the nanohole patterns on the PET film mold changed after the transfer imprinting process. The aim of this study was to experimentally verify the feasibility of this process to fabricate metal micro-disk arrays. The optical properties of the fabricated gold submicron-scale disk arrays were also investigated.

## 2. Materials and Methods

### 2.1. Silicon Wafer Mother Mold

A mother mold was made from a silicon wafer as shown in Figure 1a. The silicon wafer <100> is the best material to use as master mold. It is easy to process and allows imprinting of all kinds of micrometer patterns with a high aspect ratio. A series of micropillar patterns were fabricated using the photolithography method and dry etching techniques. Figure 1b,c present an AFM topography images and Figure 1d shows the height profile of the silicon wafer mother mold (AFM, Keyence VN-8010). The results shown that the average distance between the center of the pillars was 4000 nm, the average diameter was approximately 2000 nm, and the average height was approximately 1000 nm.

### 2.2. Thermal Embossing Process of the PET Film Mold

Figure 2a illustrates the schematic diagram of a thermal embossing machine used in this experimental work. It has a heating plate that consists of a copper plate. The upper heating plate was fixed with bolts on the mobile plate and the lower heating plate was fixed with bolts on the base plate. Ceramic plates were used to protect the components from heat exposure. The heater cartridges and thermocouples were inserted into the upper and lower heater plates and connected to the temperature control system. The digital heater temperature controller was used to control the heating temperature of the upper and lower heater plates (Digital Fine Thermo, DG2N, 100VAC-220VAC, Hakko (Thailand) Co., Ltd., Patumthani, Thailand).

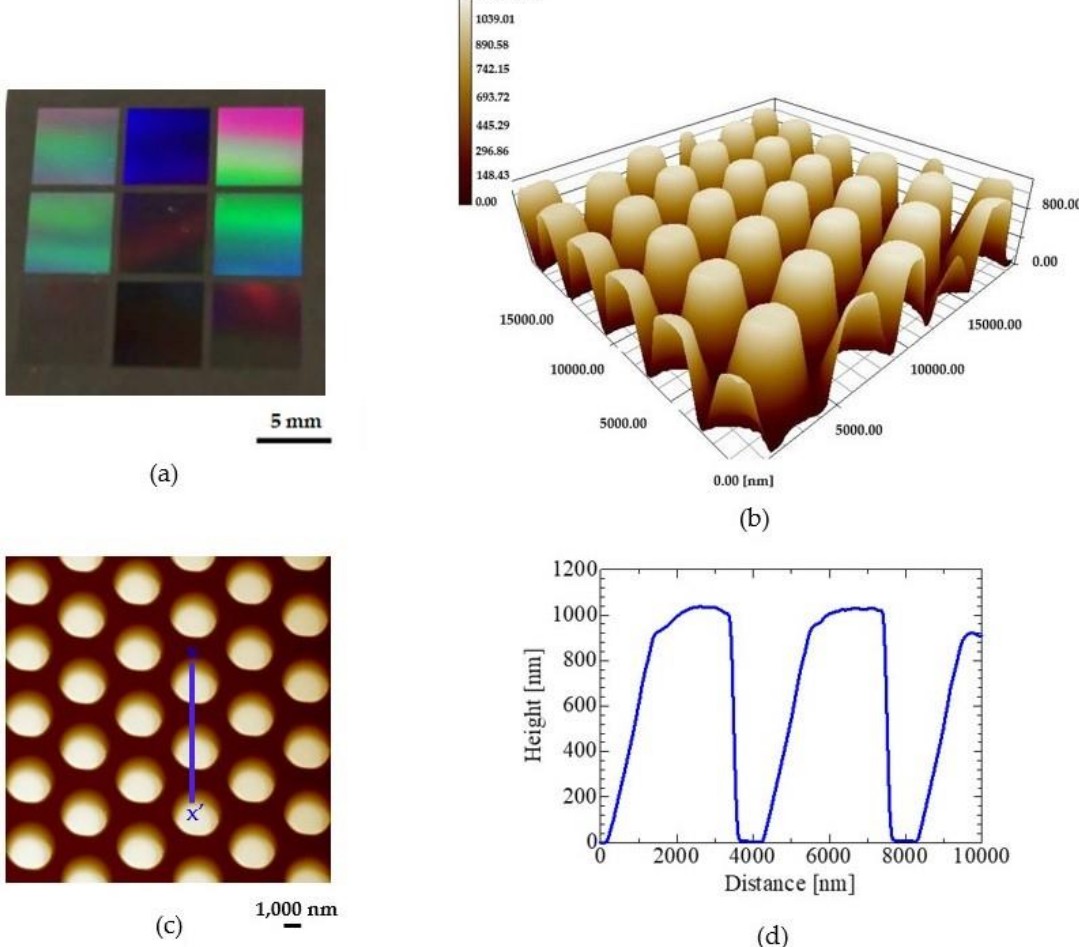

**Figure 1.** Silicon wafer mother mold; (**a**) the mother mold with nanostructures; (**b**) the three-dimensional image of the AFM topography image; (**c**) the top view of the AFM topography image; (**d**) the height profiles.

Figure 2b, step (i) shows the structure of the quartz glass plate, master mold, and PET film on the thermal embossing machine to fabricate the nanohole patterns. The PET film (LumirrorTM, S10, Toray International Trading (Thailand) Co., Ltd. (Bangkok, Thailand)) was placed under the mother mold. The size of PET film was $20 \times 20$ mm$^2$ and the thickness was 188 μm. The glass plates were placed at the top and bottom to avoid stress concentrations leading to fractures of the mother mold. The thickness of the glass plate was 1 mm with a square size of $20 \times 20$ mm$^2$. Figure 2b, step (ii) illustrates the embossing process. The weight of the upper copper plate and the moving plate was approximately 8 kg. The pressing load was approximately 0.2 MPa in the area of $20 \times 20$ mm$^2$.

Figure 3 illustrates the process curve of the thermal embossing process. In the heating phase, the assembly of a glass plate, a master mold, and a PET film was placed on the thermal embossing machine. The glass transition temperature of the PET film was about 78 °C [23]. Therefore, the build-up materials were heated from room temperature (RT) to 90 °C above the glass transition temperature. The heating time was about 4 min. In the embossing phase, a pressing load was applied by self-pressing the moving plate. The pressing time was maintained for 2 min. The flow behavior of the polymer can be improved by both temperature and pressure. Finally, in the cooling and demolding phase, the specimen was rotated out of the machine and cooled to room temperature for 3 min. Then, a PET film mold was demolded with the help of tweezers. The cycle time for the thermal embossing process was about 9 min.

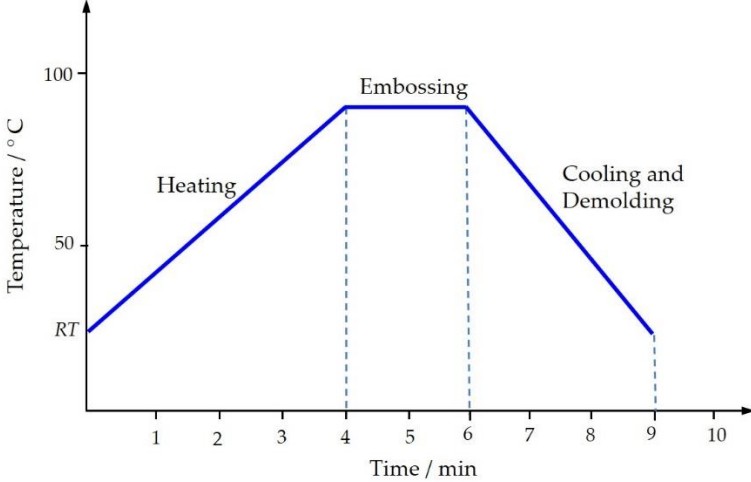

**Figure 2.** The thermal embossing method. (**a**) Schematic diagram of the in situ thermal embossing machine. (**b**) Thermal embossing process of a PET film mold.

**Figure 3.** Processing curve of the thermal embossing process to fabricate a PET film mold.

This study involves the use of nanohole patterns as a convenient and easily accessible alternative to nanoscale features fabricated via the transfer imprinting method. Figure 4a shows the PET film mold. The nanohole pattern is located in the nine square areas (25 mm$^2$) on one surface of the film PET. Figure 4b shows a three-dimensional AFM image of nanohole patterns on a PET film mold and Figure 4c shows a top view of the atomic force microscopy (AFM) topography image. The nanopatterns on a PET film mold were observed by atomic force microscopy. The height profiles of the nanohole patterned on a PET film mold had an average diameter of 2000 nm and an average depth of about 900 nm, as shown in Figure 4d. This mold was used to transfer the thin imprinting process to the gold thin film, which was coated on a substrate.

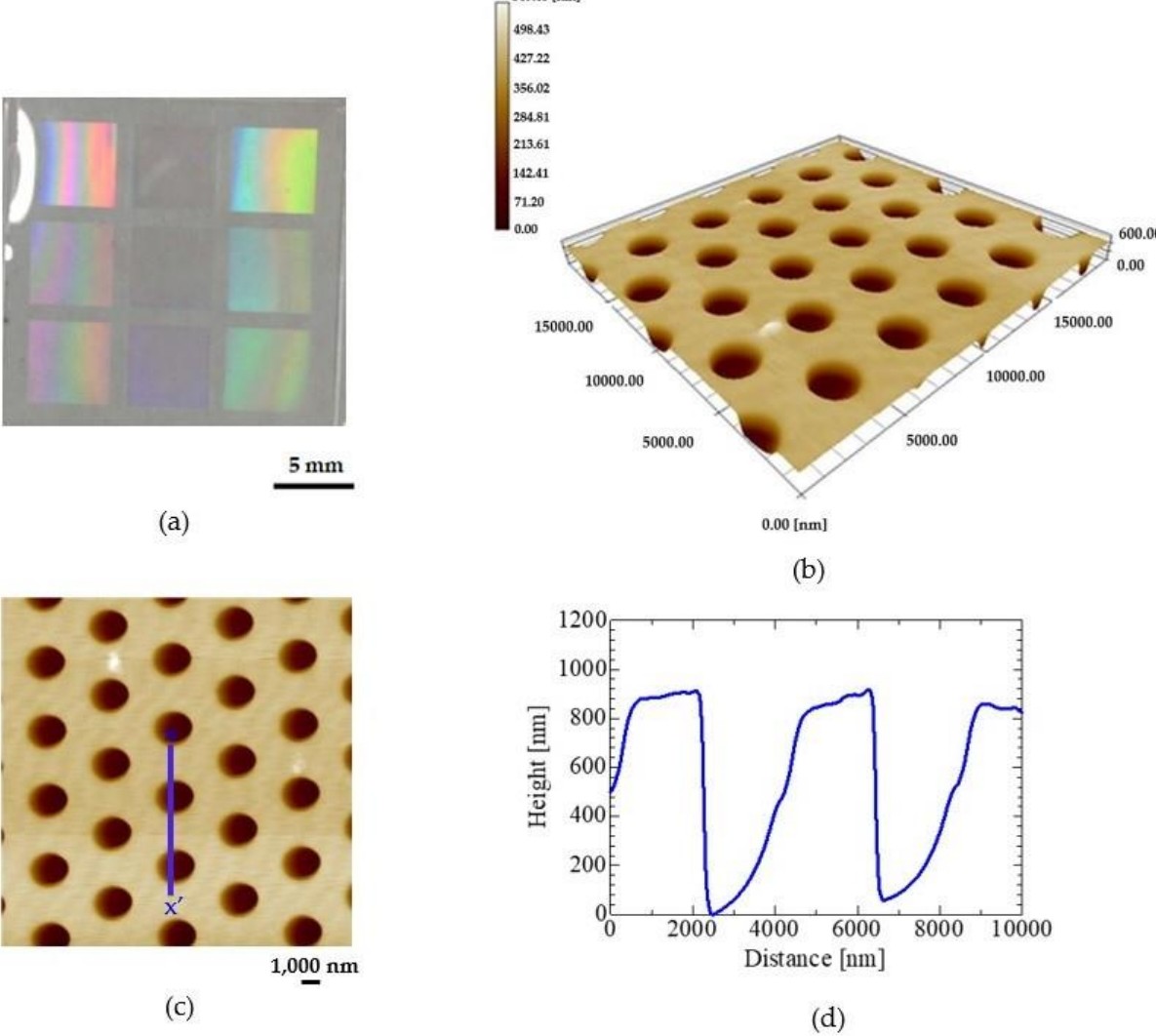

**Figure 4.** (**a**) The nanohole patterns on a PET film mold; (**b**) the three dimensional of AFM topography image; (**c**) the top view of AFM topography image; and (**d**) the height profiles.

### 2.3. Transfer Imprinting Test Using a PET Film Mold

The transfer imprinting experiment was performed according to the following procedure, which is shown in Figure 5. (i) First, a substrate was prepared from a quartz slide glass with a thickness of 1 mm and cut to a size of 12 × 12 mm$^2$. The substrate was cleaned by ultrasonic cleaning with acetone for 15 min. After the glass substrate was dried at room temperature, it was subjected to sputter etching with argon gas for 5 min to remove residual acetone molecules from the substrate surface. The pressure in the chamber of a DC sputter coater was 15 Pa, an acceleration voltage was 0.7 kV and the sputtering current was 5 mA.

The distance between the sample and the Au target on the table was 35 mm. (ii) Secondly, a thin gold film was deposited on an etched quartz glass plate with a thickness of 40 nm using a DC sputter coater. The thickness of the Au film was controlled by adjusting the sputtering time. (iii) Third, the polymer film mold was butted onto the gold thin film on a quartz glass substrate. Then, the imprinting force was added by scraping the back side of the PET film mold with tweezers. (iv) Then, the arrays of gold micro-disks were fabricated on the gold thin film. A PET film mold was pressed in only to the thin gold layer and did not damage the substrate.

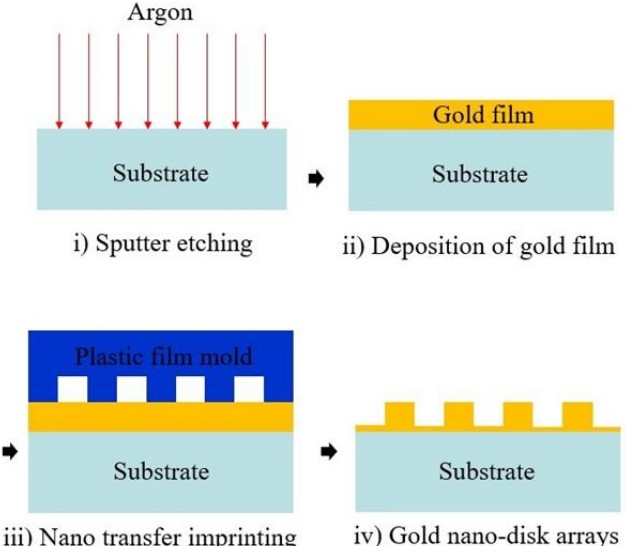

**Figure 5.** Fabrication process of metallic micro-disk arrays by mean of a PET film mold for the transfer imprinting method.

### 2.4. Localized Surface Plasmon Resonance Properties

The metallic nano-disk arrays are used for the LSPR sensing. Investigation of the optical properties of the fabricated gold micro-disk arrays on a substrate was studied. The ultraviolet-visible (UV-vis) extinction spectra were measured in an absorbance configuration using a light source with a spectrometer (Aventes, AvaSpec-ULS2048CL-EVO-UA-50). The absorbance spectra are recorded with a spectrometer in the visible light range (400–800 nm).

## 3. Results

### 3.1. Gold Microdisk Arrays on a Substrate

Figure 6a shows the gold thin film on a quartz glass substrate after the transfer imprinting experiment. The gold micro-disk arrays on a quartz glass substrate were observed by atomic force microscopy, as shown in Figure 6b. The result showed that the protuberances of the gold micro-disk arrays are regularly aligned, as shown in Figure 6c. It can be seen that the tip-to-tip distance of the gold microdisk arrays was about 4000 nm, an average diameter of about 2000 nm, and an average height of about 50 nm, as shown in Figure 6d. The figures confirm that the gold layer on a quartz glass substrate was formed into gold micro-disk arrays by the transfer imprinting process. Despite the plastic deformation of the gold thin film by the transfer imprinting process, the gold layer remained and agglomerated into the micro-disk structures on a glass substrate. Therefore, an efficient transfer imprinting method is demanded. Both the uniformity of height and the alignment of gold microdisk arrays are good.

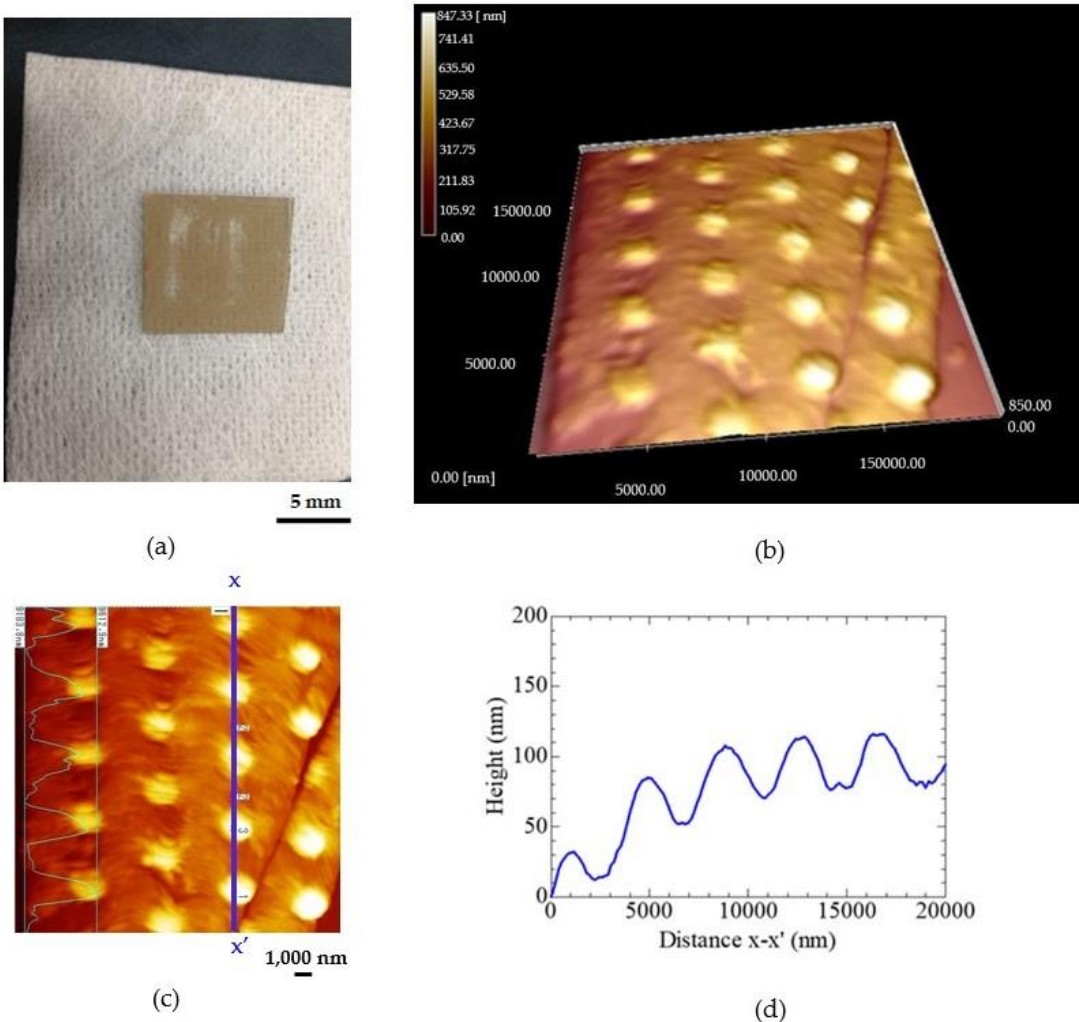

**Figure 6.** (**a**) Gold thin film on a quartz glass substrate after transfer imprinting method; (**b**) top view of the AFM topography image of the gold micro-disk arrays; (**c**) three-dimensional of the AFM topography image; (**d**) height profiles.

### 3.2. LSPR Properties of Gold Nano Disk Arrays on a Substrate

It was known that gold is often chosen for sensing applications due to its chemical stability and resistance to oxidation [24,25]. The LSPR properties of the fabricated gold micro-disk arrays on a substrate were evaluated using the absorbance spectrum. The absorbance spectra of the gold micro-disk arrays on the quartz glass substrates in the range of wavelength for 400–800 nm was studied experimentally. Figure 7 illustrates the comparison on the absorbance spectra between (i) the thin gold film with microdisk arrays on a substrate and (ii) the thickness of the gold film was 40 nm without nanostructures on a substrate. It was found from the figure that the position of the absorbance peak depends on the microstructures of gold film. For the gold micro-disk arrays fabricated on a gold film, it was found that an absorbance peak appeared on the wavelength of about 550 nm. For the gold thin film without nanostructures on a substrate, the absorbance spectrum was flat and had no pointed peak. It was confirmed that a good regularity of micro-disk arrays lead to higher enhancement, while a bare gold film causes a decrease in the peak and a broadening of the spectrum. It is possible that these enhanced absorbance spectra can be tuned by controlling the protuberance of gold nanostructure film on a substrate [26,27].

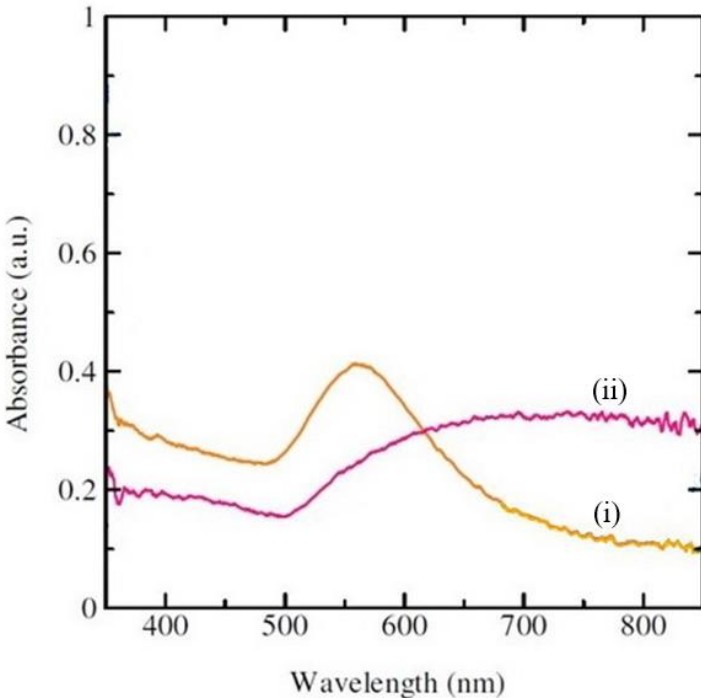

**Figure 7.** The comparison of absorbance spectrum of gold thin film between (**i**) gold thin film with micro-disk arrays and (**ii**) gold film without nanostructures, on quartz glass substrate.

## 4. Discussion

In this study, the thin gold film was deposited on a quartz glass substrate by a metal sputtering method. According to the literature, the yield stress of gold nanoparticles is approximately 216 MPa [28,29]. However, the micro-Vicker hardness test under a load of 0.05 kg was used to estimate the yield stress of the PET film mold, as shown in Figure 8. It was confirmed that the yield stress was about 45 MPa. The PET film mold was softer than the gold thin film. The PET film mold was the main tool, which was very important for the transfer imprinting process. This result shows that the gold thin film was deformed into the gold micro-disk arrays by the PET film mold.

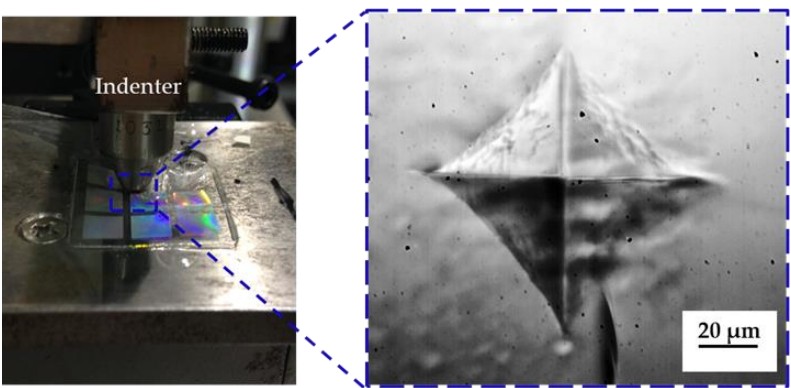

**Figure 8.** The micro-Vicker hardness testing of a PET film mold.

One of the reasons for this strange phenomenon is the voids in the gold coating layer. Figure 9 shows the schematic diagram of Au deposition on the glass substrate. The glass was placed in the chamber of a DC sputter coater, and a thin gold film was deposited on the surface of the substrate. The oxygen atom of $SiO_2$ directly interacted with an Au atom of the gold clusters ($Au_3$). Since the sputtering current of the Au coating in this experiment was quite high, it was assumed that the coated gold thin film had a coarse microstructure

in which tiny grains were piled up like a sand layer. Therefore, the true yield stress might have been much lower than that reported in the literature.

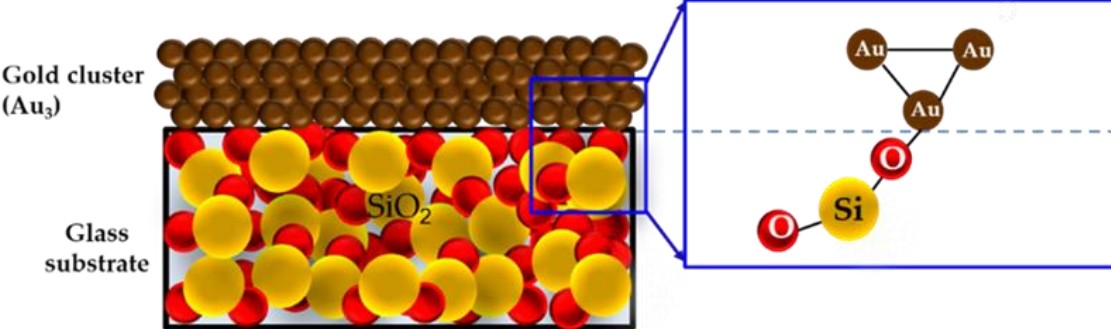

**Figure 9.** Schematic diagram of the bonding of the gold cluster atoms (Au$_3$) on a glass substrate (SiO$_2$).

Another possible reason for this phenomenon is the increase in hardness of the PET film mold caused by the pressure of the transfer imprinting. The compressive stress is concentrated in the area around a nanohole, and the stress was increased by the transfer imprinting pressure. Figure 10 illustrates a schematic diagram of the compression mechanism attributed to the micro-disk in the gold thin film on a substrate. In this study, the depth of the holes on a PET film mold was about 1000 nm. The thickness of the gold film on a quart glass substrate was 40 nm. The PET holes with the depth of 1000 nm were directly pressed onto the 40 nm-thick deposited gold film layer. After the PET holes penetrated the gold layer with a certain penetration depth, some of the gold clusters overflowed into the cavities of the PET holes. Therefore, the height of the gold layer in the cavities was continuously increased during the transfer imprinting process. It was confirmed that the gold micro-disk structures were formed with a thickness of about 50 nm on a glass substrate.

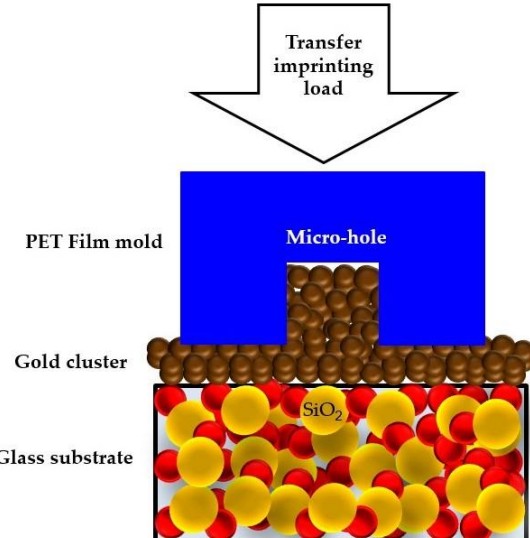

**Figure 10.** The schematic diagram of the compression mechanism for the transfer imprinting process.

The limitation of this transfer imprinting process by the PET film mold is, in fact, a reuse of the PET film mold. It was found that the dimensions of the nanohole patterns on the PET film mold changed after the transfer imprinting process. The change in the width and height of the PET film mold after use is shown in Figure 11a,b. Comparing the dimensions of the nanoholes before and after the transfer imprinting process, the average diameter of the hole expanded from 2000 nm to 2400 nm. Moreover, the average depth of the nanoholes decreased drastically from 900 nm to 200 nm. As a result, the deformation of

the nanohole patterns on the used PET film mold may affect the lower precision of the gold nanostructure on the substrate. This has the disadvantage that a PET film mold cannot be used for replication.

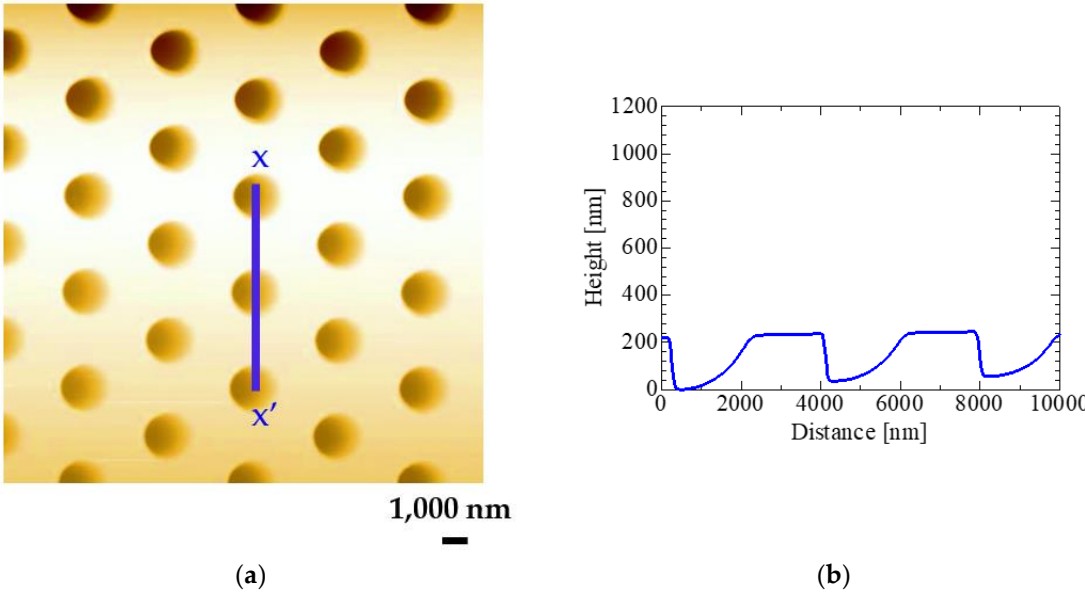

(**a**)　　　　　　　　　　　　　　　　　(**b**)

**Figure 11.** The nanohole patterns on a PET film mold after the transfer imprinting process; (**a**) the top view of AFM topography image; (**b**) the height profiles.

The advantage of this process is that it does not require expensive equipment other than the DC sputter coater. This process does not require hazardous chemicals such as an acid solution. Moreover, in this study, the soft material was used as a PET film mold. Micro- and nano-sized surface patterns were formed on the PET film by the thermal embossing process. The A4 size of the commercial PET film was fabricated to form a shape of $20 \times 20$ mm$^2$ for 155 copies. Most importantly, the PET film mold was utilized as the imprinting mold to fabricate the plasmonic nanostructures on a substrate without lithography equipment.

## 5. Conclusions

In order to develop an efficient fabrication process for metallic microstructures, a transfer imprinting process was proposed in this study. This process consists of three steps: cleaning the substrate, deposition of the metal film, and direct transfer imprinting using a PET film mold. In addition, the details of the thermal embossing process to fabricate the PET film mold were explained.

It has been shown that the gold micro-disk arrays can be fabricated by this method. The metallic films coated on the substrate were formed as ordered gold micro-disk arrays. The advantage of this process is that it does not require expensive equipment other than a sputter coater and does not require hazardous chemicals such as acidic solutions. It has also been confirmed that the minimum size of the gold micro-disk film structure formed by this process is about two micrometers in diameter. The basic mechanism of direct transfer imprinting by using a PET film mold was discussed. This was attributed to the fact that the gold clusters overflowed into the cavities of the PET holes through the transfer imprinting process. Due to the increase in hardness of the PET film form caused by the pressure of transfer imprinting process, the metal film in this area was formed into ordered gold micro disk arrays on the substrate.

The LSPR optical property of the fabricated gold micro-disk arrays were studied experimentally. As a result, it was demonstrated that LSPR is very sensitive to the pattern of submicron-scale disk arrays. The absorbance spectra of the gold thin film with micro-disk

arrays on a substrate demonstrated significant enhancement of light output characteristics at visible range wavelength. It was known that gold is often chosen for sensing applications due to its chemical stability and resistance to oxidation. Results have shown that a good regularity of micro-disk arrays led to higher enhancement. It is possible that these enhanced absorbance spectra can be tuned by controlling the protuberance of gold nanostructure film on a substrate for chemical biosensing applications.

**Funding:** The authors gratefully acknowledge the funding by the University of Phayao, Thailand.

**Institutional Review Board Statement:** Not applicable.

**Informed Consent Statement:** Not applicable.

**Data Availability Statement:** Not applicable.

**Acknowledgments:** This work was supported by the University of Phayao, Thailand.

**Conflicts of Interest:** The authors declare no conflict of interest.

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
