# Peer review of "A Study on Fabrication Process of Gold Microdisk Arrays by the Direct Imprinting Method Using a PET Film Mold"

_crystals, doi:10.3390/cryst11121452_

Round 1
Reviewer 1 Report
Comments:
In this study, the author proposes a nanofabrication process of embossing gold microdisk array using PET film mold. This process consists of thermal embossing process of the PET film mold and transfer imprinting process using a PET film mold. The tip-to-tip distance of the gold microdisk arrays was about 4 µm, an average diameter of about 2 µm, and an average height of about 50 nm in the result.
This manuscript could be modified in the following four aspects:
- The overview of embossing process.
It is necessary to introduce the advantages and disadvantages of imprinting materials of bottom-up approach in the section of introduction. It is helpful for readers to understand why the author adopts PET as the imprinting material and PET is softer than gold.
- The morphology of the gold microdisk array.
In Figure 8. (b), the height of the gold microdisk array at the distance of 1000nm is smaller than that at another position. Besides, sharp peaks appear at the top of the gold microdisk array, which is different from the platform protrusion on the PET mold. For this phenomenon, whether there is a transfer imprinting method to improve the morphology of gold microdisk array and what is the idea to improve this phenomenon.
- The explanation of the difference between the absorbance spectrums of two gold films.
It is too general to explain the absorbance peak on the wavelength of 550nm only by the morphology of gold array, which can be explained by introducing physical model or other characterization methods. Besides, there is a label error in Figure 9, the waveform (i) is for gold film without nanostructures and (ii) is for gold thin film with micro-disk arrays.
- The increasement in hardness of the PET film mold.
The PET film mold becomes harder due to the pressure of the transfer imprinting. The author is supposed to explain the above phenomenon according to references, which makes the explanation clearer and more convincing.
Author Response
Dear Reviewer,
Please see the attachment.
I sincerely thank you for your time in reviewing and providing valuable comments for my paper.
Best Regards,
POTEJANA Potejanasak (Author)

Reviewer 2 Report
In this article, an author suggested a plasmonic gold microdisk arrays by direct microimprinting. First of all, I determined that there is a big difference between the subject (all aspects of Crystallography) covered in this journal and the content of the submitted manuscript. I decided that it would be better to re-submit it to a journal that is highly related to optical elements and manufacturing processes.
Also, I summarized some points that need to be supplemented as follows.
(1) the quality of the photos and pictures constituting each figure should be improved.
(2) There is a typo in the figure description, which requires sufficient consideration and correction. For example, Figure 9. ~ (ii) gold film without nanostructures, on quart glass substrate. 'quart' should be corrected to 'quartz'.
(3) I think Figures 2 to 4 can be combined into one figure.
(4) The researcher presented absorbance spectrum data of the developed device, and a sufficient supplementary explanation on how it can be applied is needed in a section of Conclusion.
(5) Advanced techniques for nanoscale imprints have been developed and published by various research groups (for instance, AIP Advances 7(12), 125125 (2017), Nanomaterials, 9(11), 1519 (2019), ), so merits and needs should be fully explained when compared to these technologies.
Author Response

(The authors gave the same response as above.)

Reviewer 3 Report
Reviewer Report on the Submitted Manuscript
Title: A study on fabrication process of gold microdisk arrays by the direct imprinting method using a PET film mold
Special Issue: 3D Printing/Additive Manufacturing of Polymeric Nanostructures with Multifunction Properties
First, the submitted manuscript is based on the conventional nanoprinting method. I am sorry that it is difficult to judge the present topic is fitted well the special issue or not. Straightly saying, it is not fitted the special issue. The present topic should be submitted to regular issue.
Author Response

(The authors gave the same response as above.)

Reviewer 4 Report
This work describes the nanofabrication process of embossing gold microdisk array using a PET film mold.
In general, the manuscript is well structured, written, and complete.
I propose some improvements that will help the manuscript:
- Have an English review to correct some typo errors and readability. (i.e. line 54: “ex-ample”; Figure 9, “quart”; and label confusion?)
- Improve the quality of the figures (Figure 1, 7, and 8). There is not good sharpness.
- Too many figures. Perhaps some can be merged (Figure 2, 3), (Figure 4, 5).
- The description of Figure 9 in section 3.2 is too generic. Are there any references that could support the explanation of the absorption differences?
Author Response

(The authors gave the same response as above.)

Round 2
Reviewer 2 Report
As suggested in the previous review, I think that there is a big difference between the subject (all aspects of Crystallography) covered in this journal and the content of the submitted manuscript. So, I maintained my opinion of rejecting this paper and it would be better to submit the manuscript to a journal that is highly related to fabricate nano-components. Also, the quality improvement in Figure 10 is also considered necessary, and Figure 11 does not indicate what profile it is showing.
Author Response
Dear Reviewer,
Please see the attachment.
Best regards,
POTEJANA P.

Reviewer 3 Report
2nd Round Reviewer Report on the Submitted Manuscript Entitled: A study on fabrication process of gold microdisk arrays by the direct imprinting method using a PET film mold
Special Issue "3D Printing/Additive Manufacturing of Alloys, Ceramics and Polymers"
Now the content of the topic is fitted to the content of the above special issue.
Authors measured all topography images of silicon wafer mother mold with a height of 1000 nm and replica of PET film with a height of 900 nm using an atomic force microscope (AFM). AFM is a simple and easy way to characterize the surface, but authors should know the limitation of the depth resolution of AFM. Topographical image of the surface with a couple of nanometer and tens nanometer sizes is well characterized using an AFM tip, but huge depth close to 1 micron is not well characterized by an AFM tip. Reviewer also had experienced above trouble in the past. In that case, we usually use the cross section image of SEM to characterize the depth profile of holes. In this meaning, artificial images are reproduced in Figures 1 (d) and others. The inside wall of the silicon mother mold should be like sharp cliff almost 90 degrees. In other words, even though hole is cylinder shape, we will measure the depth profile such as in figure 1 (d). In Figure 1 (d), for example, the cross section profile should be diameter of 2000 nm and neighboring distance (bottom distance) also should be 2000 nm. Reviewer suggests that all cross section profiles should be checked by a SEM or other way. In this meaning, in Figure 6 (d), the depth profile of the replicated PET is not correctly measured and we cannot judge how the replica of PET is well reproduced. This is a serious problem for the present manuscript. Anyway, all cross section profiles and images should be reinvestigated.
Minor question is the order of structures author fabricated is micrometer order. The resolution may be couple of tens or hundreds nanometer order. For example, in Figure 4 or Figure 11, can you say PET nanohole? The diameter of hole is 2000 nm order.
Author Response
Dear Reviewer,
Thank you so much for your suggestions.
"Please see the attachment.
Best regards,
Author
